# The Addition of Glutamine Enhances the Quality of Huangjiu by Modifying the Assembly and Metabolic Activities of Microorganisms during the Fermentation Process

**DOI:** 10.3390/foods13172833

**Published:** 2024-09-06

**Authors:** Jiajia Jiang, Guanyu Fang, Changling Wu, Peng Wang, Yongzhu Zhang, Cheng Zhang, Fenghua Wu, Zhichu Shan, Qingru Liu, Xingquan Liu

**Affiliations:** 1College of Forestry and Biotechnology, Zhejiang Agriculture and Forest University, Hangzhou 311300, China; 2020102112002@stu.zafu.edu.cn; 2College of Food and Health, Zhejiang Agriculture and Forest University, Hangzhou 311300, China; wuchangling0805@163.com (C.W.); wpeng@zafu.edu.cn (P.W.); zhangyz@zafu.edu.cn (Y.Z.); zhangcc@zafu.edu.cn (C.Z.); 3National Grain Industry (High-Quality Rice Storage in Temperate and Humid Region) Technology Innovation Center, Zhejiang Agriculture and Forest University, Hangzhou 311300, China; wufh@zafu.edu.cn; 4College of Advanced Agricultural Sciences, Zhejiang Agriculture and Forest University, Hangzhou 311300, China; 5Zhejiang Pagoda Brand Shaoxing Rice Wine Co., Ltd., Shaoxing 312000, China; tpszc@163.com; 6Food Microbiology Key Laboratory of Sichuan Province, Xihua University, Chengdu 610039, China; seaflyru@126.com

**Keywords:** huangjiu, volatile components, amino acids, quality improvement, fermentation

## Abstract

In this study, the effects of adding glutamate (Glu), glutamine (Gln), aspartate (Asp), and asparagine (Asn) on the flavor formation of Huangjiu were investigated, and the effect of Gln concentration on the quality, microbial community structure, and flavor development of Huangjiu was further explored. Varied Gln concentrations influenced yeast growth, sugar utilization, microbial communities, and quality attributes. Additional Gln promoted yeast cell counts and sugar depletion. It increased the complexity of bacterial co-occurrence networks and reduced the impact of stochastic processes on assembly. Correlation analysis linked microorganisms to flavor compounds. Isolation experiments verified the role of *Saccharomyces cerevisiae*, *Aspergillus chevalieri*, *Bacillus altitudinis*, and *Lactobacillus coryniformis* in flavor production under Gln conditions. This research elucidated the microbiological mechanisms by which amino acid supplementation, especially Gln, enhances Huangjiu quality by modulating microbial metabolic functions and community dynamics during fermentation. This research is significant for guiding the production of Huangjiu and enhancing its quality.

## 1. Introduction

Huangjiu is one of the oldest alcoholic beverages in China. Produced predominantly through the fermentation of steamed polished glutinous rice, the complex sensory profile of Huangjiu is revered for its intricate balance of flavors and aromas fundamentally attributed to the microbiological activity within the fermentative ecosystem. The *Saccharomyces* species, the workhorse microbes of alcoholic fermentation, are accompanied by a cadre of other fungi and bacteria, each contributing uniquely to the nuances of the final product [1]. The quest to optimize and control the quality of Huangjiu has, therefore, long hinged on an understanding of these microbial interactions and their influence on fermentation.

Amino acids are vital for microbial metabolism and, hence, for the fermentation process. Nitrogen sources are important nutrients for microbial growth and metabolism, and the metabolism of microorganisms is an important source of flavor compounds in wine. Amino acids act as important nitrogen sources; changing the composition of amino acids in the fermentation medium can profoundly affect microbial metabolism, influencing Huangjiu’s flavor, aroma, and especially the generation of higher alcohols and esters and overall quality [2,3]. Yeast assimilates amino acids through highly regulated transport systems. This assimilation and metabolism directly impact yeast cell growth, viability, and fermentation kinetics [4]. However, the relationship between amino acid availability and the production of secondary metabolites, such as higher alcohols and esters that contribute to the complex aroma of Huangjiu, is intricate [3]. The undertaking of these interactions is crucial, as they represent the basis for modulating fermentation processes to enhance wine quality. For instance, certain amino acids act as precursors for volatile aroma compounds synthesized by yeast during fermentation [5]. Thus, studying the specificity of amino acid utilization by yeast strains used in Huangjiu production will contribute to a better understanding of flavor development. The sulfur-containing amino acids, such as methionine and cysteine, are another area of interest due to their dual role in yeast metabolism and the potential for contributing to off-flavors through the generation of volatile sulfides under certain fermentation conditions [6]. Therefore, understanding the regulation of microbial growth and metabolism by the addition of amino acids is the key to analyzing their impact on the quality of Huangjiu. 

Additionally, nitrogen availability can shape the interactions among microorganisms within a community [7]. Cooperative and competitive interactions occur between microbial taxa, and the type and concentration of nitrogen sources can influence the balance between these interactions. Some microorganisms may rely on the metabolic by-products of other nitrogen-utilizing species, leading to mutualistic interactions [2]. Conversely, competition for limited nitrogen resources can result in antagonistic interactions, where certain organisms may inhibit the growth of others. Exploring the effects of nitrogen sources on microbial interactions is essential for comprehending community dynamics and predicting fermentation outcomes. Moreover, nitrogen availability can influence the assembly processes of microbial communities during fermentation [2]. Microbial community assembly refers to the processes governing the recruitment and establishment of microorganisms within a given ecosystem. Nitrogen sources can act as selective pressures, shaping the recruitment and successional patterns of microbial populations [8]. For example, studies have demonstrated that certain nitrogen sources can enhance the colonization and growth of certain microbial species, leading to their dominance in the community over time [9]. Understanding the impact of nitrogen sources on community assembly is vital for designing strategies to manipulate microbial populations and improve fermentation efficiency. Amino acids are an important nitrogen source in the fermentation process of Huangjiu. However, it is currently unknown what kind of impact this nitrogen source will have on the microbial community structure, and thus on the quality of Huangjiu. It is of great significance to use nitrogen sources to regulate the quality of Huangjiu by changing the nitrogen source composition in the fermentation environment by adding exogenous amino acids to explore the microbiological mechanism of the effect of amino acid composition on the quality of rice wine. 

This study investigated the effects of adding different amino acids on the quality of Huangjiu and further explored the impact of varying Gln concentrations in the fermentation liquid on the microbial community structure and flavor of Huangjiu. In addition, through correlation analysis, we deciphered the underlying microbiological mechanisms of how amino acid addition affects the quality of Huangjiu and verified these findings with fermentation experiments. This research is significant for guiding the production of Huangjiu and enhancing its quality.

## 2. Materials and Methods

### 2.1. Sample Preparation and Collection

The method of Huangjiu brewing is based on a previous study [10]. After soaking 500 g of glutinous rice for two days, the rice was steamed and allowed to cool. Then, 500 mL of water, 50 g of wheat Qu, and 1 g of yeast were added to initiate fermentation. To investigate the effects of adding different amino acids on the quality of yellow wine during fermentation, four amino acids (Glu, Gln, Asp, Asn) that can be preferentially utilized by yeast were selected as additional nitrogen sources and added at a concentration of 10 mmol/L before the start of fermentation [11,12]. The group without added amino acids was used as a control (CK), with 6 parallel groups in each group. To investigate the effect of adding different concentrations of Gln on the quality of Huangjiu, we set up three experiments: a control group (CK), a fermentation broth with Gln added at a concentration of 200 mg/L group (Gln-200), and a fermentation broth with Gln added at a concentration of 400 mg/L group (Gln-400). Before sampling, we stirred the fermentation broth evenly, and took 6 parallel samples from each group.

### 2.2. Physicochemical Parameters, Flavor Substances, Free Amino Acid Detection

The physicochemical parameters of Huangjiu samples were detected according to a previous study [10]. The volatile compounds were detected using headspace solid-phase microextraction combined with gas chromatography–mass spectrometry (HS-SPME/GC-MS) [13,14]. The initial temperature was 40 °C, held for 5 min, then increased to 120 °C at a rate of 6 °C/min, held for 5 min. Finally, the temperature was increased from 120 to 190 °C at a rate of 3 °C/min, held for 5 min. The following MS parameters were selected: EI source, electron energy of 70 eV, full scan mode to collect data, ion source temperature of 230 °C, interface temperature of 250 °C, scanning range of 35–500 *m*/*z* (mass-to-charge ratio), and solvent removal time of 3 min. According to a previous study of our term, high-performance liquid chromatography (HPLC) was used to determine the free amino acid content in Huangjiu samples with ChromCore TM C18 column (250 mm × 4.6 mm × 5 μm) [10]. 

### 2.3. DAN Extraction, PCR, and Amplicon Sequencing Analysis

Based on the number of yeast cells during fermentation process of Huangjiu (Figure 1A), the DNA (deoxyribonucleic acid) of 6 replicates of wine samples from Day 3 control and experimental group were extracted by CTAB method. The ITS region in the fungal genes was amplified using the primer ITS3-2024F (5′-GCATCGATGAAGAACGCAGC-3′)/ITS4-2409R (5′-TCCTCCGCTTATTGATATGC-3′), and the V3–V4 region of the bacterial 16S r RNA (16S ribosomal ribonucleic acid) gene was amplified using primer sets 341F (5′-CCTAYGGGRBGCASCAG-3′)/806R (5′-GGACTACNNGGGTATCTAAT-3′). The amplified products were sequenced in Beijing Novaseq Technology Co., Ltd. (Beijing, China). Raw tags were obtained from reads via FLASH v1.2.11. Clean tags were then generated using fastap, and the effective tags were further generated by removing the chimeras using Vsearch. The effective tags were denoised and filtrated into amplicon sequencing variants (ASVs) via QIIME2, and the representative sequences were annotated against the SILVA v138 and UNITE v8.2 database.

### 2.4. Microbial Isolation and Fermentation Experiment

PDA (Potato Dextrose Agar) and MRS (De Man, Rogosa and Sharpe Agar, with 2% CaCO_3_) media were used for microbial isolation. To determine the effect of Gln addition on the metabolism of these microorganisms, we conducted fermentation experiments. A total of 100 g of glutinous rice flour was liquefied at 80 °C for 2 h by adding 20 mg α-amylase and 2 L water; then, we used lactate, which regulated pH to 4.3. After that, we added 80 mg amyloglucosidase, and the saccharification solution was obtained by saccharification at 60 °C for 2 h. A total of 100 mL saccharification solution and 5 ml microbe solution/spore suspension was taken into 250 mL Erlenmeyer flasks. Gln was added at a concentration of 0, 200, 400, or 800 mg/L. After these solutions were sealed, they were cultured at 30 °C for 5 days, and the volatile flavor changes were detected by HS-SPME-GC/MS, the changes in yeast and bacteria cell density were characterized by the absorbance at 600 nm, and the fungal growth differences were indicated by the dry weight of mycelium.

### 2.5. Statistical Analysis

Analysis was carried out in triplicate at least for each sample. IBM SPSS Statistics 26 was used for significance (one-way ANOVA and Duncan test) and Spearman correlation analysis. The final results were presented as mean ± standard deviation (SD), with different letters indicating significant differences at the 0.05 level. Spearman correction only showed the results with significant correlations of |*r*| > 0.7, *p* < 0.05, and the thickness of the line indicates the strength of the correlation. Gephi v0.9.2 was used for visualization of co-occurrence networks [15]. The R package of microeco v0.9.0 was used for analysis related to microbial communities [16].

## 3. Results and Discussion

### 3.1. Effect of Amino Acid Addition on Physicochemical Properties of Huangjiu

The metabolism of amino acids and other nitrogenous compounds by yeast can significantly affect the quality of the wine. Existing research indicates that the four amino acids Glu, Gln, Asp, and Asn are preferred by yeast. Therefore, the impact of the addition of these four amino acids on the quality of Huangjiu was investigated (Figure 1). The results show that the addition of four amino acids caused a significant increase in the pH of Huangjiu (*p* < 0.05) (Figure 1E). Besides pH, the addition of Gln did not have a significant effect on the other physicochemical indicators of Huangjiu (*p* > 0.05). The addition of Glu and Asp significantly reduced the content of titratable acid in Huangjiu, with its content dropping from 7.87 ± 0.43 g/L in the CK group to 7.04 ± 0.29 and 7.18 ± 0.25 g/L, respectively (Figure 1B). Additionally, the addition of Asp (0.55 ± 0.01 g/L) and Asn (0.54 ± 0.02 g/L) can significantly increase the content of amino acid nitrogen in Huangjiu, while the addition of Glu (0.46 ± 0.01 g/L) significantly reduces the content of amino acid nitrogen in Huangjiu (CK: 0.49 ± 0.02 g/L) (Figure 1C). Compared to the CK group (ethanol: 8.1 ± 0.2%vol, total sugar: 18.55 ± 3.14 g/L), the addition of Asp and Asn can, respectively, increase the total sugar content and ethanol in Huangjiu to 9.2 ± 0.2%vol and 28.88 ± 0.85 g/L (Figure 1A,D).

### 3.2. Effect of Amino Acid Addition on Content of Volatile Components of Huangjiu 

HS-SPME/GC-MS was used to determine the content of volatile flavor components in Huangjiu after fermentation. A total of 53 volatile components were detected in the Huangjiu samples, including 28 esters, 18 alcohols, 4 acids, and 3 aldehydes (Appendix A). The detected substances in Huangjiu were predominantly composed of esters and alcohols, both in terms of concentration and diversity. This observation aligns with prior research, confirming that the principal volatile constituents of Huangjiu are indeed esters and alcohols [1,17]. These compounds are largely responsible for the aromatic and flavor profiles of Huangjiu [18,19]. 

After the addition of Glu, Gln, Asp, and Asn to Huangjiu, the number of volatile components increased from 35 in the control group (CK) to 39, 39, 36, and 44, respectively. In addition, the addition of different amino acids has a significant impact on the content of esters and alcohols in Huangjiu. The Huangjiu industry’s development has been notably obstructed by the issue of consumers getting intoxicated quickly, which has been a key factor impacting the intake of alcoholic beverages [3]. The content of higher alcohols is generally high in Huangjiu, which are the main compounds that accelerate intoxication from Huangjiu [3]. Therefore, a comparison was made on the content of higher alcohols in each group. The addition of Gln and Asn significantly reduces the total alcohol content in Huangjiu, primarily by decreasing the levels of higher alcohols. Isoamyl alcohol is the predominant higher alcohol found in all Huangjiu samples. The addition of Gln to Huangjiu during the fermentation process can significantly decrease the content of volatile compounds, with reductions of isoamyl alcohol and phenylethyl alcohol by 26.89% and 45.45%, respectively, compared to the control group (CK). Over recent years, evidence has accumulated suggesting that the higher alcohols in Huangjiu are more intoxicating than those in brandy, red wine, rum, whiskey, or vodka, resulting in more pronounced negative physiological effects from its consumption [20]. Incorporating Gln into the Huangjiu fermentation process can markedly decrease the levels of higher alcohols, thus improving the quality of the Huangjiu. Therefore, the following research has concentrated on examining how varying glutamine concentrations during fermentation impact the microbial composition and flavor profile of Huangjiu.

The addition of different amino acids preferred by yeast has different effects on the content of free amino acids in fermented Huangjiu. The addition of yeast-preferred amino acids significantly increased the content of His, Leu, and Lys in Huangjiu, while the content of the yeast-preferred amino acids Asp and Glu significantly decreased (Appendix A). This indicates that adding preferred amino acids significantly improved the utilization of the yeast’s preferred amino acids Asp and Glu, reduced the utilization rate of non-preferred amino acids, and resulted in a higher content of non-preferred amino acids than in the control. This may be due to the inhibition of the non-preferred nitrogen source metabolism by the addition of preferred nitrogen sources [21]. The addition of Asp and Glu significantly increased the total free amino acid content, while the addition of other amino acids had no significant effect on the amino acid content in yellow wine. Amino acids are important precursors of higher alcohols, and they can produce corresponding higher alcohols through the Ehrlich pathway.

### 3.3. Effect of Gln Addition on Physicochemical Properties of Huangjiu 

The experimental results show that incorporating Gln into the fermentation process of Huangjiu can effectively reduce the concentration of higher alcohols in the Huangjiu, thereby improving its quality without disrupting the normal fermentation process. Thus, we conducted a study to examine the effects of various concentrations of Gln on the microbial community structure and the flavor development throughout the fermentation of Huangjiu. Yeasts are the key microorganism in the fermentation process of Huangjiu; therefore, the cell count of the yeasts was monitored during the fermentation process. With the onset of fermentation, there was a significant augmentation in the population of yeast cells, culminating in a peak on the third day (CK: 5.84 ± 0.88 × 10^8^ cells/mL, Gln-200: 6.70 ± 0.50 × 10^8^ cells/mL, and Gln-400: 6.31 ± 0.44 × 10^8^ cells/mL) (Figure 2A). The number of yeast cells in the Gln-200 group was significantly higher than that in the CK group (*p* < 0.05), while there was no significant difference between the Gln-400 and CK group (*p* > 0.05). During the late stage of fermentation, the number of yeast cells in the Gln-200 group was 1.75 ± 0.22 × 10^8^ cells/mL, showing a significant increase of 28.57% compared to the CK group (1.25 ± 0.35 × 10^8^ cells/mL) (*p* < 0.05). This result highlights the significant growth-promoting effect of adding Gln to the yeasts.

The variation trend in the total sugar content serves as a crucial indicator of the Huangjiu fermentation process [22]. In addition, the total sugar content is an important indicator for evaluating the quality of Huangjiu [23]. Therefore, the total sugar content was monitored during the fermentation process of Huangjiu with the addition of different concentrations of Gln (Figure 2B). The results showed that the addition of Gln significantly enhanced the utilization rate and efficiency of the total sugar during the Huangjiu fermentation process. After one day of fermentation, the total sugar content in the Huangjiu of the Gln-400 group decreased rapidly (31.02 ± 6.39 g/L), significantly lower than that in the CK group (59.01 ± 11.52 g/L) and the Gln-200 group (52.48 ± 5.04 g/L) (*p* < 0.01). After 2 days of fermentation, the total sugar content in the Huangjiu of both the Gln-200 (14.10 ± 1.15 g/L) and Gln-400 (12.01 ± 2.50 g/L) groups was significantly lower than that of the CK group (31.02 ± 6.39 g/L), and this state was consistently maintained thereafter (*p* < 0.01). This indicates that the addition of Gln promoted yeast growth, therefore improving the utilization rate of total sugars in the fermentation liquid.

Moreover, the physicochemical properties of Huangjiu after fermentation were detected. The addition of Gln can significantly reduce the levels of titratable acid (CK: 7.28 ± 0.37 g/L, Gln-200: 4.93 ± 0.23 g/L, and Gln-400: 6.45 ± 0.35 g/L) and total sugar (CK: 20.56 ± 5.11 g/L, Gln-200: 3.10 ± 1.23 g/L, and Gln-400: 1.94 ± 0.71 g/L) in Huangjiu (*p* < 0.05), but it can increase the pH (CK: 3.97 ± 0.08, Gln-200: 4.27 ± 0.07, and Gln-400: 4.20 ± 0.03) and amino-acid-bound nitrogen content (CK: 0.43 ± 0.01 g/L, Gln-200: 0.47 ± 0.03 g/L, and Gln-400: 0.46 ± 0.02 g/L) of Huangjiu (*p* < 0.05) (Figure 2C–E,G). However, contrary to expectations, there was no significant difference in ethanol content between the groups (CK: 9.40 ± 1.80%vol, Gln-200: 8.90 ± 1.56%vol, and Gln-400: 9.23 ± 0.98%vol) (*p* > 0.05) (Figure 2F).

### 3.4. Analysis of Microbial Community Structure 

Microorganisms play a crucial role in determining the quality of fermented foods, as their growth and metabolism yield an abundance of flavor compounds and nutritional elements. To elucidate the microbiological underpinnings of how Gln supplementation influences the quality of Huangjiu, a detailed assessment of the microbial community dynamics was performed throughout the Huangjiu fermentation process. The results reveal that after the addition of Gln, there was no significant change in the Chao1 index (CK: 165.37 ± 74.87, Gln-200: 219.50 ± 122.74, and Gln-400: 220.93 ± 130.71) of the bacteria in the Huangjiu (*p* > 0.05), while the Shannon index (CK: 2.39 ± 0.20, Gln-200: 2.99 ± 0.62, and Gln-400: 2.69 ± 0.43) increased significantly (*p* < 0.05) (Figure 3A). Non-metric multidimensional scaling (NMDS) was performed to evaluate the differences between bacterial communities in each group based on the unweighted UniFrac distance (Figure 3B). The result showed that there is a lack of clear differentiation among the samples from each group within the NMDS analysis. This indicated that the structures of bacterial communities across the groups show no significant variations. To further resolve the differences in bacterial community structure between the groups, the permutational multivariate analysis (PerMANOVA) of variance was performed [24]. There are no significant differences in the bacterial community structures between the groups (PerMANOVA, *p* > 0.05).

Bacteria from the phyla Firmicutes and Proteobacteria dominate, with their relative abundances reaching 63.96 ± 16.85% and 32.46 ± 17.48%, respectively (Appendix A). In addition, bacteria belonging to the phyla Actinobacteriota were detected in all Huangjiu samples. The predominant bacterial genera include Weissella (average relative abundance: 47.42 ± 20.30%), Enterobacter (11.09 ± 12.53%), Lactococcus (9.20 ± 9.57%), Pantoea (6.89 ± 4.78%), Bacillus (1.29 ± 1.35%), Lactobacillus (1.25 ± 4.77%), Saccharopolyspora (1.14 ± 1.09%), etc. (Figure 3C). Weissella, as a genus of the lactic acid bacteria, is frequently detected in fermented foods and can enhance the flavor of fermented products through its metabolism [25,26]. The ternary phase diagram suggests an enrichment pattern where Lactobacillus, Lactococcus, and Weissella were predominantly found in the Gln-200 group and Kosakonia, Saccharopolyspora, and Escherichia-Shigella showed significant presence in the CK group.

Fungal communities were detected during the Huangjiu fermentation after the addition of Gln. The results showed that after adding Gln, there was no significant change in the Chao1 index (CK: 79.69 ± 8.51, Gln-200: 66.11 ± 57.34, and Gln-400: 63.63 ± 23.88) of the fungal community in the Huangjiu (*p* > 0.05), while the average value of the Chao1 index decreased. After the addition of Gln, there was a significant decrease in the Shannon index (CK: 1.31 ± 0.22, Gln-200: 1.03 ± 0.20, and Gln-400: 1.00 ± 0.15) of the fungal community in the Huangjiu (*p* < 0.05) (Figure 3D). These results indicate that the addition of Gln reduces the diversity of the fungal community during the fermentation process of Huangjiu. The beta diversity of the fungal community indicated that the there were no significant differences in the structure of the fungal communities between the different groups, and the result of PerMANOVA displayed the same result (*p* > 0.05) (Figure 3E). 

In all samples of Huangjiu, fungi from the Ascomycota phylum showed undisputed predominance, with their average relative abundance surpassing 80% (Appendix A). Saccharomyces (73.57 ± 8.63%) and Aspergillus (3.81 ± 4.88%) are the two genera with the highest relative abundance in the Huangjiu samples (Figure 3F). The fungi from these two genera play a crucial role in the fermentation process of Huangjiu [27,28]. Aspergillus can convert starch into reducing sugars through its metabolic processes, and Saccharomyces can utilize these reducing sugars to produce ethanol. Furthermore, fungi from these two genera are also significant contributors to the flavor profile of Huangjiu. The ternary phase diagram suggests an enrichment pattern where Aspergillus (CK: 2.87 ± 4.98%, Gln-200: 2.35 ± 0.64%, and Gln-400: 2.25 ± 0.70%) is predominantly found in the CK group, Plectosphaerella (CK: 0, Gln-200: 0.022 ± 0.050%, and Gln-400: 0) shows a significant presence in the Gln-200 group, and Talaromyces (CK: 0.018 ± 0.017%, Gln-200: 0.010 ± 0.017%, and Gln-400: 0.67 ± 1.56%) is notably concentrated in the Gln-400 group.

### 3.5. Analysis of the Microbial Community Assembly Process 

The construction of cooccurrence networks for the ASVs within the microbial community was carried out across distinct groups. Network analysis could offer a comprehensive and distinct insight into microbial relationships and ecological assembly principles, going beyond just understanding the structure and bio-diversity of microbial communities [29]. The microbial co-occurrence network is an analytical method used to study the structure and interactions of microbial communities. In microbiology and ecology, microbial co-occurrence networks can help us understand the relationships between different microbial species and how they coexist and interact within ecosystems [30]. The respective influence of stochastic factors in the composition of microbial community groups was investigated through the neutral theory model. The ecological steps that shape microbial community development become apparent upon analyzing the changes in phylogenetic relationships among similar organisms, which is vital to differentiating between deterministic, niche-related and stochastic, neutral influences in community assembly [31,32]. Deterministic processes result from ecological selection imposed by biotic and abiotic factors, which predictably filter species, affecting the fitness of organisms and thus determining the composition and relative abundance of species [33]. Stochastic processes involve random births, deaths, probabilistic dispersal, and the random fluctuation of species’ relative abundances (ecological drift), and are not the result of fitness determined by the environment [34]. The results of the neutral theory model indicate that the random factors affecting microbial assembly during the fermentation process of Huangjiu are reduced after the addition of Gln (Figure 4A,B). The explanatory power (R^2^) of the neutral theory model for bacterial communities decreased from 0.35 in the CK group to 0.02 in the Gln-200 group and 0.13 in the Gln-400 group (Figure 4A). Similarly, the addition of Gln also reduced the impact of stochastic processes on fungal community assembly. The R^2^ of the neutral theory model for fungal communities decreased from 0.50 in the CK group to 0.35 in the Gln-200 group and 0.25 in the Gln-400 group (Figure 4B). These results indicate that the addition of Gln increases the impact of deterministic processes on microbial community assembly.

The co-occurrence networks of the bacterial community were constructed for different groups, and the results indicated that the addition of Gln increased the complexity of the bacterial co-occurrence network. The average degree of the co-occurrence network increased from 5.81 in the CK group to 69.48 in the Gln-200 group and 19.92 in the Gln-400 group (Figure 4C). In addition, the modularity of the co-occurrence network decreased from 0.69 in the CK group to 0.31 in the Gln-200 group and 0.55 in the Gln-400 group. The increase in complexity of the bacterial co-occurrence network suggests an intensification of interactions between microbes. A previous study has indicated that the enhanced complexity of microbial interactions can significantly improve the multifunctionality of flavor compound production in food fermentation micro-ecosystems [35]. Microorganisms can produce flavor compounds in fermented foods through metabolic complementation. The analysis results of constructing co-occurrence networks for fungal communities indicate that the addition of Gln does not have a noticeable effect on the co-occurrence network of fungal communities (Figure 4D). The average degree of co-occurrence networks for the CK, Gln-200, and Gln-400 groups were 3.48, 4.55, and 2.67, respectively. The modularity of co-occurrence networks for the CK, Gln-200, and Gln-400 groups were 0.61, 0.61, and 0.82, respectively. This is consistent with previous research indicating that fungal communities exhibit greater robustness when faced with environmental selection pressures [36].

### 3.6. Correlation Analysis between Microorganisms and Flavor Substances

The above results indicate that the addition of Gln significantly changes the assembly and interaction patterns of microorganisms. The interaction between microorganisms will significantly change the metabolic functions of microbial communities [37]. The content of volatile components and amino acids in Huangjiu was detected (Appendix A). To elucidate the microbiological mechanisms impacting the quality of Huangjiu after Gln supplementation, bacterial and fungal genera exhibiting a relative abundance exceeding 1% were chosen for correlation analysis with flavor compounds (Figure 5A). A total of nine genera exhibited significant correlations with flavor compounds (*p* < 0.05, r > 0.7, Spearman correlation). A total of eight genera of microorganisms were found to have a significant correlation with flavor substances, including two fungal genera (Saccharomyces and Aspergillus) and six bacterial genera (Bacillus, Pantoea, Weissella, Saccharopolyspora, Lactococcus, and Enterobacter). Seven genera of microorganisms were found to have a significant positive correlation with flavor compounds. Pantoea and Saccharomyces both displayed a significant positive correlation with 13 types of flavor substances, Lactococcus with 12 types, Weissella with 10 types, Bacillus with 8 types, and both Aspergillus and Enterobacter with 1 type each. Microorganisms from seven genera exhibited significant negative correlations with flavor substances. Weissella was negatively correlated with ten flavor substances, Lactococcus with eight, Aspergillus, Pantoea, and Saccharomyces each show negative correlations with five flavor substances, Saccharopolyspora with three, and Bacillus with two.

Lactic acid bacteria were significantly correlated with ester compounds. Among them, Lactococcus showed significant negative correlations with ethyl lactate, ethyl palmitate, methyl oleate, and ethyl linoleate, while Weissella exhibited significant positive correlations with ethyl lactate and ethyl palmitate (*p* < 0.05). Lactic acid bacteria were the main producers of lactic acid in Huangjiu, and ethyl lactate, which is formed through the esterification reaction of lactic acid and ethanol, is an important flavor compound in Huangjiu [1]. Therefore, it can be inferred that the decrease in lactic acid bacteria caused by Gln addition was the reason for the decrease in ethyl lactate. Ethyl palmitate is the main ester substance in Huangjiu [38]. Isoamyl alcohol, isobutanol, and phenylethyl alcohol are the main alcohols in Huangjiu and are typical representatives of higher alcohols [3]. Saccharomyces is considered the primary microorganism responsible for alcohol production and can produce large amounts of higher alcohols through the Ehrlich and Harris pathways [39,40]. In this study, isobutanol, isoamyl alcohol, and phenylethyl alcohol are significantly positively correlated with Saccharomyces. In addition, previous research has shown that higher alcohols are significantly positively correlated with Saccharomyces [41,42]. These studies highlight the key role of Saccharomyces in the production of higher alcohols. Isoamyl alcohol was the higher alcohol with the highest content in Huangjiu and shows the strongest positive correlation with Saccharomyces. However, it should be noted that the formation of isoamyl alcohol is the result of multiple microbial metabolisms [43]. Moreover, isoamyl alcohol is significantly negatively correlated with Weissella and notably positively correlated with Lactococcus and Pantoea. Saccharomyces and Aspergillus each had significant correlations with three distinct free amino acids. In the bacterial genus, Bacillus, Pantoea, Weissella, Mitochondria, and Saccharopolyspora exhibited significant correlations with seven, four, four, four, and three free amino acids, respectively. Bacillus exhibited significant positive correlations with Gly, Pro, Val, Asp, Ile, and Lys, and showed a significant negative correlation with Cys. These results were in agreement with a previous study that found that the addition of Bacillus had increased the amino acid content in fermented food [44]. 

### 3.7. Experimental Verification of the Effect of Gln Addition on the Flavor of Huangjiu

The above research results showed that the addition of Gln significantly affected the structure, interaction patterns, and assembly process of microbial communities, thereby significantly affecting the quality of Huangjiu. In order to analyze the microbiological mechanisms of the effect of Gln addition and fermentation on the flavor of Huangjiu, microorganisms were isolated during the fermentation process of Huangjiu, and 1 *Saccharomyces cerevisiae*, 1 *Aspergillus chevalieri*, 1 *Bacillus altitudinis*, and 1 *Lactobacillus coryniformis* were obtained (Figure 5B). Then the flavor compounds of these four strains fermented in saccharification solution were determined. Saccharomyces is the primary aroma-producing microorganism in the fermentation process of Huangjiu, with the ethanol generated from the yeast’s alcoholic fermentation providing a significant amount of precursor for the creation of ester flavor components in the wine. The fermentation experiments with *Saccharomyces cerevisiae* demonstrated that the addition of Gln significantly impacted the production of flavor compounds in the *Saccharomyces cerevisiae* (Figure 5C). The incorporation of Gln notably decreased the levels of higher alcohols like isobutanol, isoamyl alcohol, and phenylethanol in the fermentation broth. This reduction coheres with the lowered higher alcohol content found in Huangjiu upon Gln supplementation, suggesting that the influence of Gln on the metabolism of Saccharomyces plays a vital role in significantly reducing the concentration of higher alcohols in Huangjiu. The results of correlation analysis indicate a significant positive correlation between 4-vinylguaiacol and Saccharomyces. In the fermentation experiment of *Saccharomyces cerevisiae*, it was shown that the addition of Gln increased the production of 4-vinylguaiacol in the fermentation broth. 4-Vinylguaiacol is an important aroma component in alcoholic beverages, with the aroma of cloves and wheat beer [45]. The fermentation experiment results of *Aspergillus chevalieri* showed that the addition of Gln promoted the growth of *Aspergillus chevalieri* and reduced the content of Ethanol, 2- (dodecyloxy)—in the fermentation broth, which is consistent with the significant negative correlation between Aspergillus and Ethanol, 2-(dodecyloxy)—(Figure 5C). The fermentation experiment of *Bacillus altitudinis* showed that the content of ethyl octanoic acid, ethyl ester; decanoic acid, ethyl ester; pentadecanoic acid, ethyl ester; and octadecanoic acid, ethyl ester was relatively high in the control group. After the addition of Gln, the ability of *Bacillus altitudinis* to generate ethyl esters significantly decreased, and this indicates that Bacillus may be a microorganism that reduces the content of ethyl compounds in Huangjiu after adding Gln (Figure 5C). Moreover, the fermentation experiment of Lactobacillus corniformis showed that the addition of Gln promoted the growth of Lactobacillus corniformis and increased the organic acid content in Huangjiu. This is consistent with the significant positive correlation between lactic acid bacteria and organic acids in Huangjiu.

## 4. Conclusions

This study investigated the effects of glutamine addition on the microbial communities and product quality of Huangjiu fermentation. Results showed that glutamine supplementation altered the structure and interactions of bacterial and fungal communities. It increased the complexity of bacterial co-occurrence networks and reduced the impact of stochastic processes on community assembly. Correlation analysis revealed that *Saccharomyces cerevisiae*, *Aspergillus chevalieri*, *Bacillus*, etc., were linked to flavor compounds. Experimental verification confirmed the effects of dominant microbes on flavor formation. In conclusion, glutamine addition influenced the quality of Huangjiu by remodeling microbial communities and their relationships during fermentation.

## Figures and Tables

**Figure 1 foods-13-02833-f001:**
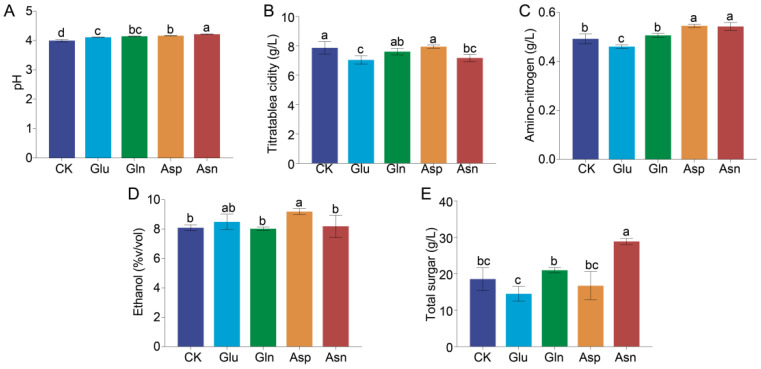
Physicochemical parameters of Huangjiu fermented with different amino acid additions. The pH (**A**), titratable acid content (**B**), amino-nitrogen content (**C**), ethanol concentration (**D**), and total sugar content (**E**), of Huangjiu. Different letters represent significant differences (*p* < 0.05).

**Figure 2 foods-13-02833-f002:**
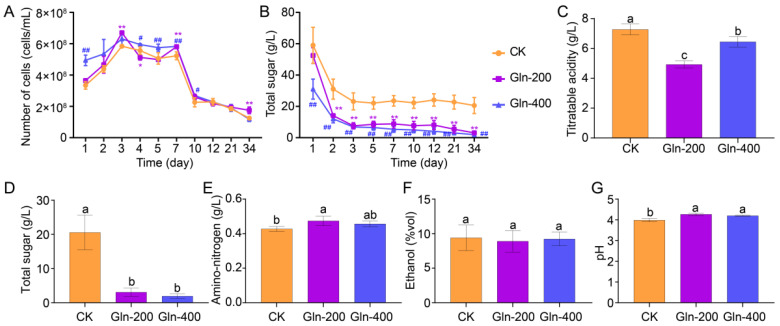
The effect of adding different concentrations of Gln fermentation on the quality of Huangjiu. The number of yeast cells (**A**) and total sugar content (**B**) during the fermentation process. #, *p* < 0.05, ##, *p* < 0.005, *, *p* < 0.05, **, *p* < 0.005. Titratable acid content (**C**), total sugar content (**D**), amino-nitrogen content (**E**), ethanol concentration (**F**), and pH (**G**) of the Huangjiu samples. Different letters represent significant differences (*p* < 0.05).

**Figure 3 foods-13-02833-f003:**
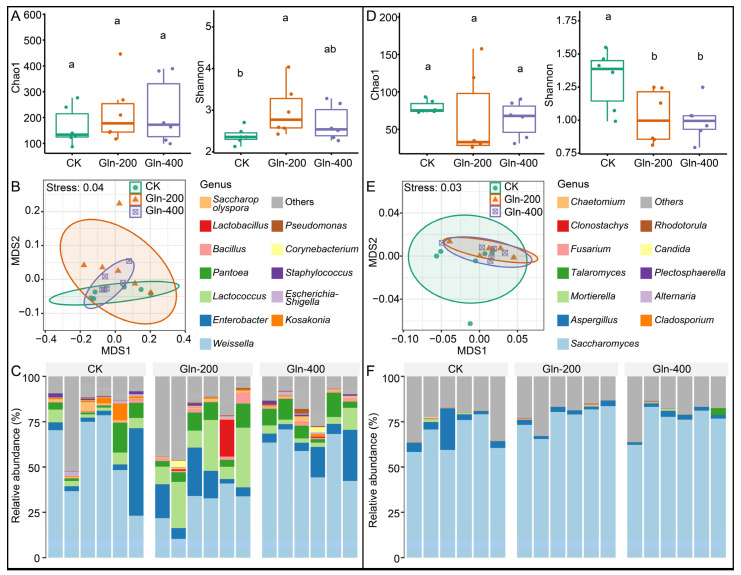
The effect of adding different concentrations of Gln on the microbial community during the fermentation process of Huangjiu. The alpha (**A**), beta diversity (**B**), and structure (**C**) of the bacterial community. The alpha (**D**), beta deversity (**E**), and structure (**F**) of the fungal community. Different letters represent significant differences (*p* < 0.05).

**Figure 4 foods-13-02833-f004:**
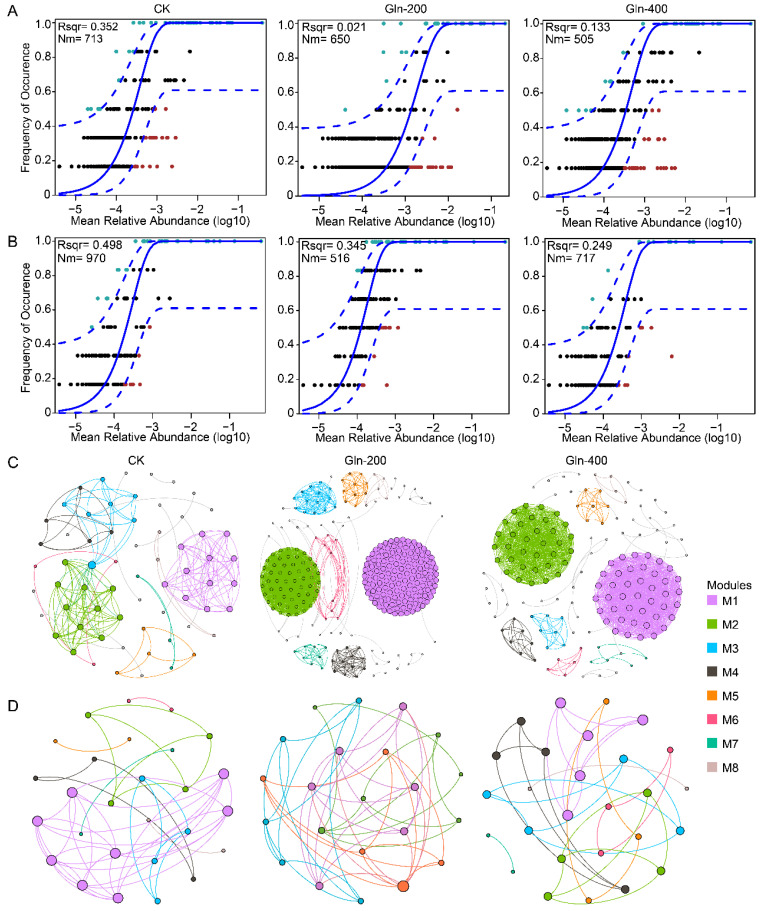
Fit of the neutral community model (NCM) of community assembly of bacteria (**A**) and fungi (**B**). The solid blue lines indicate the best fit to the NCM, and the dashed blue lines represent 95% confidence intervals around the model prediction. Nm indicates the metacommunity size times immigration, and R^2^ indicates the fit to this model. The black dots represent the ASVs predicted by the model, the red dots represent ASVs with frequencies lower than the predicted ones, and the blue dots represent ASVs with frequencies higher than the predicted ones. Co-occurrence networks of the bacterial (**C**) and fungal community (**D**), and the M1–8 represents the modules in the networks.

**Figure 5 foods-13-02833-f005:**
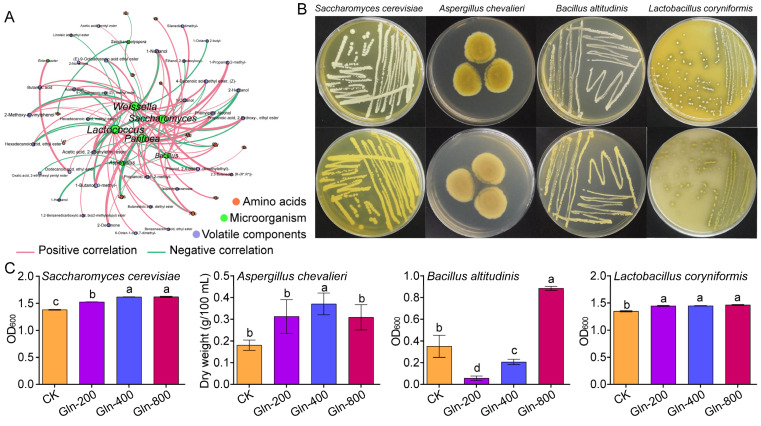
(**A**) Correlation analysis between microorganisms and flavor substances. (**B**) Image of microbial colonies isolated from Huangjiu. (**C**) Biomass of *Saccharomyces cerevisiae*, *Aspergillus chevalieri*, *Bacillus altitudinis*, and *Lactobacillus coryniformis* in fermentation broth. No Gln was added to CK, and 200, 400, 800 indicates different amounts of Gln. Different letters represent significant differences (*p* < 0.05).

## Data Availability

The original contributions presented in the study are included in the article/Appendix A, further inquiries can be directed to the corresponding authors.

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
