# Peer review of "The Addition of Glutamine Enhances the Quality of Huangjiu by Modifying the Assembly and Metabolic Activities of Microorganisms during the Fermentation Process"

_foods, 2024, doi:10.3390/foods13172833_

Round 1

Reviewer 1 Report

Comments and Suggestions for Authors

Author Response

Comment 1: Give the names of different amino acids in the abstract. Only glutamine is presented

there.

Resonse 1: Thanks for your advice. All the name of different amino acids has been added in abstract. You can find the change in page 1, line 17-20.

Comment 2: In 2.1 Sample preparation and collection: Provide information and site the literature sources explaining why you have chosen to study in particular these amino acids: Glu, Gln, Asp and Asn.

Resonse 2: Thanks for your advice, provide information and literature sources have been added in 2.1. You can find the change in page 3, line 101-103.

Comment 3: There are abbreviations, that are not explained in the text, such as DAN, PCR, DNA, PDA, MRS, etc. Include explanation for all abbreviations in the text.

Resonse 3: Thanks for your advice, We have added an explanation of all abbreviations. You can find the change in page 3, line 125, 129, 138-139.

Comment 4:How many days does the fermentation take and why the samples are taken from Day 3 /Please see 2.3 DAN extraction, PCR, and amplicon sequencing analysis /?

Resonse 4: The fermentation take 34 days, and according to Figure 1A, the number of yeast cells was the highest on Day3, so the samples from Day3 were selected for DAN extraction, PCR, and amplicon sequencing analysis. You can find the change in page 3, line 124-126.

Comment 5:You should clearly explain for which experiments you added 10 mmol/L, 200 mg/L, 400 mg/L and 800 mg/L Gln. /See section 2.1 and 2.4. also Figure 5/.

Resonse 5: First, we selected 10mmol/L as the addition amount of four amino acids to explore the effects of different amino acid additions on the fermentation of Huangjiu, then selected Gln as the external nitrogen source to explore the effects of different addition concentrations (including 200, 400 and 800 mg/L) of Gin on fermentation, and finally isolated the single colonies of different microorganisms to explore the effects of different addition concentrations of Gin (including 200, 400 and 800 mg/L) on the flavor generation of purebred microorganisms to verify their flavor function. These are described in 2.1 and 2.4, at the same time, it is also further explained in the notes in Figure 5. You can find the change in page 3, line 101-103, page 12, line 443-447.

Comment 6:The labels in Figure 1 are unreadable. They must be zoomed. Also, the numbering A), B), C), D), E) is missing under the graphs.

Resonse 6: Thanks for your suggestion, we have revised it as your suggestion. You can find the change in page 4, line 161-165.

Comment 7:For clarity when you explain the results make tables or refer to the tables in the

supplementary materials, providing their names.

Resonse 7: Thanks for your suggestion, we have revised it as your suggestion. You can find the change in page 5, line 185, 216-217, page 7, line 293, page 8, line 317, page 10, line 396.

Comment 8:Provide the current names of supplementary materials when you site them in the text.

Resonse 8: Thanks for your suggestion, we have revised it as your suggestion.You can find the change in page 5, line 185, 216-217, page 7, line 293, page 8, line 317, page 10, line 396.

Comment 9:There is wrong numbering in Figure 2 /two figures are named with D/.

Resonse 9: Thank you for pointing out the mistakes,we have corrected the mistakes. You can find the change in page 6, line 227-231.

Comment 10:It is more appropriate to place Figure 3 after section 3.4.

Resonse 10: Thanks for your suggestion, we have revised it as your suggestion. You can find the change in page 8, line 330-334.

Comment 11:Zoom Figure 5 for elucidating the text.

Resonse 11: Thanks for your suggestion, we have revised it as your suggestion.You can find the change in page 12, line 442-447.

Reviewer 2 Report

Comments and Suggestions for Authors

Comments to the article “Addition of glutamine enhances the quality of Huangjiu by 2

modifying the assembly and metabolic activities of microorganisms during the fermentation process” is an interesting proposal, abording topics from the use of nitrogen sources, biological disposables such as amino acids, and its relation with growth and compound production

From introduction, a restructuration due to the information must be repetitive is recommended to edit some sections:

line 41 is recommended to be unified with the next idea as one.

Line 44-46, rewrite clearly, describing how the nitrogen source affects overgrowth, metabolic pathways depending on the nitrogen source, and homogeneous and non-homogenous communities of microorganisms. It could be possible to regulate growth only with amino acid addition.

Line 70-80, edition from the paragraph is necessary because to appears some references are missing.

Line 81 is not clear the aim of the investigation; according to the title, this paragraph must be edited

From methodology

Lines 90-101 is not clear the mean of parallel samples, and parallel groups

Line 127, how you be sure to have an appropriate description of the population from isolates, how many strains from each plate were selected, and under what criteria.

Line 135 shows that there are no solids in the sample, despite the changes in cell density. How were they removed?

From results

Letters from the figure are missing A to E

Line 179 describes a diminution in the high alcohol concentrations, but it does not describe what happens with them. They are used as precursors from other compounds or simply are missing.

Line 276 and posterior do not correct the use of microbes. It is recommended that the term microorganisms or bacteria be used for this paragraph. This term is used posterior to this mention. Please edit.

Line 346, please edit ntworks.

Conclusion

It is not necessary to discuss results; it is recommended that only the correlation of yeast and bacteria over aromatic compounds and the glutamine addition be discussed.

Author Response

Comment 1: line 41 is recommended to be unified with the next idea as one.

Resonse 1: Thanks for your advice. But the first paragraph mainly explains the influence of microorganisms on the quality of Huangjiu, and the latter paragraph mainly highlights the influence of amino acids on the microbial community, We believe that the subparagraph would better highlight the focus of both.

Comment 2: Line 44-46, rewrite clearly, describing how the nitrogen source affects overgrowth, metabolic pathways depending on the nitrogen source, and homogeneous and non-homogenous communities of microorganisms. It could be possible to regulate growth only with amino acid addition.

Resonse 2: Thanks for your suggestion, we have revised it as your suggestion. You can find the change in page 1-2, line 43-49.

Comment 3: Line 70-80, edition from the paragraph is necessary because to appears some references are missing.

Resonse 3: Thanks for your suggestion, we have added more references. You can find the change in page 2, line 74-76.

Comment 4:Line 81 is not clear the aim of the investigation; according to the title, this paragraph must be edited

Resonse 4: Thanks for your suggestion, we have revised it as your suggestion. You can find the change in page 2, line 81-87.

Comment 5:Lines 90-101 is not clear the mean of parallel samples, and parallel groups

Resonse 5: To investigate the effect of adding different concentrations Gln on the quality of Huangjiu, here we conducted two experiments. First, we selected 10mmol/L as the addition amount of four amino acids to explore the effects of different amino acid additions on the fermentation of Huangjiu. Then selected Gln as the external nitrogen source to explore the effects of different addition concentrations (including 200, 400 and 800 mg/L) of Gin on fermentation. In our manuscripts we use ’’To investigate the effect of adding different concentrations Gln on the quality of Huangjiu’’ to illustrate.

Comment 6:Line 127, how you be sure to have an appropriate description of the population from isolates, how many strains from each plate were selected, and under what criteria.

Resonse 6: Microbiome sequencing is done by the company, and we only provide samples from Day3. The isolation of microorganisms was carried out by plate isolation and purification, and by comparing the characteristics of the community, different single colonies were selected for multiple scribing and separation, and then sent to the company for colony identification. It is specified in the ‘Materials and Methods’ section.

Comment 7:Line 135 shows that there are no solids in the sample, despite the changes in cell density. How were they removed?

Resonse 7: Glutinous rice flour undergoes the action of α-amylase and amyloglucosidase to form a saccharified solution, which only takes the supernatant when used for microbial fermentation, so there is no solids in the sample.

Comment 8:Letters from the figure are missing A to E

Resonse 8: Thank you for pointing out the mistakes, we have corrected the mistakes. You can find the change in page 4, line 161-165.

Comment 9: Line 179 describes a diminution in the high alcohol concentrations, but it does not describe what happens with them. They are used as precursors from other compounds or simply are missing.

Resonse 9: The high content of higher alcohols in rice wine is the main reason of rapid intoxication in Huangjiu, and we found that the content of higher alcohols decreased after the addition of amino acids, which showed that the adjustment of amino acid composition had a good effect on reducing the content of higher alcohols, and these are specified in the second paragraph of 3.2.

Comment 10: Line 276 and posterior do not correct the use of microbes. It is recommended that the term microorganisms or bacteria be used for this paragraph. This term is used posterior to this mention. Please edit.

Resonse 10: Thanks for your suggestion, we have revised it as your suggestion. You can find the change in page 7, line 292, 294.

Comment 11:Line 346, please edit networks.

Resonse 11: Thanks for your suggestion, we have revised it as your suggestion. You can find the change in page 12, line 442.

Comment 12:It is not necessary to discuss results; it is recommended that only the correlation of yeast and bacteria over aromatic compounds and the glutamine addition be discussed.

Resonse 12: Thanks for your suggestion, but we believe that the main purpose of this study is to explain what differences in microbial metabolism caused by amino acid addition affect the changes in flavor quality of Huangjiu, so we summarize the relevant conclusions. The discussion of correlation is explained in detail in the ‘Results and Discussion ’ section.

Round 2

Reviewer 2 Report

Comments and Suggestions for Authors

Thanks for corrections

Best regards
